# Crosstalk between DNA Damage Repair and Metabolic Regulation in Hematopoietic Stem Cells

**DOI:** 10.3390/cells13090733

**Published:** 2024-04-24

**Authors:** Jian Xu, Peiwen Fei, Dennis W. Simon, Michael J. Morowitz, Parinda A. Mehta, Wei Du

**Affiliations:** 1Division of Hematology and Oncology, School of Medicine, University of Pittsburgh, Pittsburgh, PA 15232, USA; 2UPMC Hillman Cancer Center, Pittsburgh, PA 15213, USA; 3Cancer Biology, University of Hawaii Cancer Center, University of Hawaii, Honolulu, HI 96812, USA; 4Department of Critical Care Medicine, School of Medicine, University of Pittsburgh, Pittsburgh, PA 15213, USA; 5Department of Surgery, School of Medicine, University of Pittsburgh, Pittsburgh, PA 15213, USA; 6Division of Blood and Marrow Transplantation and Immune Deficiency, Cincinnati Children’s Hospital Medical Center, Cincinnati, OH 45229, USA

**Keywords:** hematopoietic stem cells (HSCs), DNA damage repair (DDR), cellular metabolism, Fanconi anemia (FA) pathway

## Abstract

Self-renewal and differentiation are two characteristics of hematopoietic stem cells (HSCs). Under steady physiological conditions, most primitive HSCs remain quiescent in the bone marrow (BM). They respond to different stimuli to refresh the blood system. The transition from quiescence to activation is accompanied by major changes in metabolism, a fundamental cellular process in living organisms that produces or consumes energy. Cellular metabolism is now considered to be a key regulator of HSC maintenance. Interestingly, HSCs possess a distinct metabolic profile with a preference for glycolysis rather than oxidative phosphorylation (OXPHOS) for energy production. Byproducts from the cellular metabolism can also damage DNA. To counteract such insults, mammalian cells have evolved a complex and efficient DNA damage repair (DDR) system to eliminate various DNA lesions and guard genomic stability. Given the enormous regenerative potential coupled with the lifetime persistence of HSCs, tight control of HSC genome stability is essential. The intersection of DDR and the HSC metabolism has recently emerged as an area of intense research interest, unraveling the profound connections between genomic stability and cellular energetics. In this brief review, we delve into the interplay between DDR deficiency and the metabolic reprogramming of HSCs, shedding light on the dynamic relationship that governs the fate and functionality of these remarkable stem cells. Understanding the crosstalk between DDR and the cellular metabolism will open a new avenue of research designed to target these interacting pathways for improving HSC function and treating hematologic disorders.

## 1. Introduction

Hematopoietic stem cells (HSCs) are a rare population of cells residing in the bone marrow (BM) niche, where they preserve the capacity to self-renew and differentiate [1,2]. HSCs are quiescent under a steady state. In response to stress, HSCs mobilize out of the niche, entering the cell cycle for division [3]. The transition from quiescence to activation is accompanied by major metabolic and mitochondrial changes that are important for balanced decisions between self-renewal and differentiation to generate enough hematopoietic stem progenitor cells (HSPCs) while preventing HSC exhaustion. Indeed, the regulation of HSC maintenance is significantly influenced by the cellular metabolism. In contrast to other differentiated cells, HSCs primarily depend on anaerobic glycolysis to generate energy. As HSCs differentiate, there is a transition towards mitochondrial oxidative phosphorylation (OXPHOS). Interestingly, aberrations in energy metabolism have been implicated as a potential mechanism contributing to the HSC abnormalities observed in diverse hematological disorders, such as BM failure and leukemia [4].

Fundamental cellular processes involved in metabolism can also damage DNA through increasing reactive oxygen species (ROSs) or generating toxic byproducts. It has emerged that cellular metabolic regulation not only generates DNA damage but also impacts DNA repair. In fact, metabolic enzymes and metabolites represent a key group of factors within the DNA damage repair (DDR) system [5]. The observation that cancer cells exhibit alterations in DDR coupled with modifications in cellular metabolism further emphasizes the links between these two fundamental processes.

One of the best studied hematologic disease models is Fanconi anemia (FA), a genetic disorder associated with BM failure, a progression to myelodysplastic syndrome (MDS) and acute myeloid leukemia (AML). FA is genetically heterogenous and plays a significant role in the DDR process [6]. Recent studies highlight the impact of FA deficiency on energy metabolism in HSCs through suppressing glycolysis and favoring the OXPHOS mediated by the p53-synthesis of cytochrome c oxidase 2 (SCO2) and the p53-TP53-induced glycolysis regulator (TIGAR) axis; and enhancing the fatty acid oxidation (FAO) mediated by the FANCD1/HES1/PPARγ axis [7,8,9,10], paving the way for explaining the intricate interplay between DDR and HSC metabolic reprogramming. In this review, we delve into the metabolic processes of HSCs, including glycolysis, OXPHOS, lipid metabolism, and amino acid catabolism, exploring how these metabolic pathways interact with DDR machinery to sustain the biological function of HSCs.

## 2. Metabolic Choreography in HSCs

Cellular metabolism, defined as the sum of the biochemical processes that produce or consume energy [11], is at the foundation of all biological activities in living organisms [12]. Core metabolism can be divided into the pathways involving abundant nutrients like carbohydrates, fatty acids, and amino acids, essential for energy homeostasis and macromolecular synthesis in humans. During the commitment to cell proliferation, extensive metabolic rewiring occurs to acquire sufficient amounts of nutrients necessary to support cell growth and to deal with the redox challenges that arise from the increased metabolic activity associated with anabolic processes [12]. Nestled within the BM niche, HSCs boast a unique metabolic profile that distinguishes them from their differentiated progenies. As outlined in Figure 1, glycolysis and the tricarboxylic acid (TCA) cycle in HSCs are indirectly connected via other nutrient pathways, such as fatty acid and amino acid metabolism, which generate acetyl-CoA and provide it to the TCA cycle. Cellular metabolism has now emerged as a key regulator of HSC maintenance [13,14]. These distinctions are elaborated below.

### 2.1. Glycolysis and Mitochondrial Oxidative Phosphorylation

As the site of respiration/oxidative phosphorylation, mitochondria, the highly dynamic organelles, provide an efficient route for the cells to generate ATP from energy-rich molecules. The role of the mitochondria in ATP generation is dependent on ambient oxygen tension [13,15,16]. Under normal oxygen tensions, cells catabolize glucose to pyruvate. Pyruvate is then imported into the mitochondria for further catabolism through the Krebs cycle, which transfers electrons to the respiratory chain for ATP synthesis. In low-oxygen conditions, ATP levels are primarily generated via glycolysis, where the intermediate metabolite pyruvate is shunted to the cytosol to be further converted to lactate and nicotinamide adenine dinucleotide+ (NAD+), continuously driving the glycolytic process [17].

HSCs residing in the BM, rely on glycolysis rather than more efficient mitochondrial OXPHOS for energy production, which aligns with the fact that HSCs primarily remain in a quiescent state in the hypoxic BM microenvironment [7]. One direct advantage of this relative dependence is correlated to the reduction of mitochondrial ROSs, which aids in various signaling pathways necessary for normal HSC functions at physiological levels [18,19]. In fact, the enriched glycolysis-dependent pathway along with a reduced oxidative capacity and lower mitochondrial activity have been proposed as markers of stemness [18,20,21,22,23].

Mechanistically, many important regulatory pathways, such as MEIS1/HIF1a, MYC, PPM1K/CDC20, and ROS signals, are known to fine-tune the nutrient metabolisms and cell fate commitments in HSCs [24]. For example, HSCs highly express pyruvate dehydrogenase kinase (Pdk), a glycolytic enzyme that inhibits the pyruvate dehydrogenase (PDH)-mediated conversion of pyruvate to acetyl-CoA, to restrict aerobic metabolism [19]. In HSCs, the elevated levels of Pdk result in the active suppression of the influx of glycolytic metabolites into the mitochondria. This glycolytic metabolic status is a cell cycle checkpoint that modulates HSC quiescence and stemness [19]. The transcription factor, hypoxia-inducible factor 1-alpha (HIF1α), is also highly expressed in long-term HSCs, mediating cellular and systemic responses to reduced oxygen availability through anaerobic-biased energy metabolism rather than OXPHOS, promoting HSC maintenance by reducing ROS production [13,25,26]. In addition, the TSC-mTOR (Tuberous sclerosis complex-mammalian target of rapamycin mTOR) pathway maintains HSC quiescence through the repression of mitochondrial biogenesis, therefore reducing ROS production, and is implicated in leukemogenesis [27].

While glycolysis is critical for stem cell maintenance, mitochondrial OXPHOS is required for stem cell differentiation [13]. During HSC differentiation and maturation, a rapid switch from glycolysis to mitochondrial OXPHOS and ATP generation occurs [19,25,28,29,30]. This switch is accompanied by an elevation of ROSs, mTOR activation, enhanced mitochondrial biogenesis, and increased oxygen consumption [14], allowing the fast-cycling differentiating cells to meet their altered and higher metabolic and energy needs associated with differentiation [20,28,31]. Indeed, the ATP and ROS levels in lineage-committed progenitors are much higher than those in HSCs [18,32]. Previous study reported a differentiation block of PTPMT1, a PTEN-like mitochondrial phosphatase-depleted HSCs [20].

### 2.2. Fatty Acid Metabolism

Lipids, essential nutrients in the body, serve as a source of energy storage and metabolism, and participate in the regulation of multiple crucial biological processes [33]. Lipids encompass a diverse range, including fatty acids, triglycerides, phospholipids, and cholesterol, all of which play various vital roles in the organism, such as constructing cell membranes, serving as important molecules for energy storage and transfer, and cell signal transduction.

As a major component of lipid metabolism, fatty acids undergo an oxidative breakdown in the mitochondria, releasing energy for the organism’s activities. Fatty acid oxidation is a primary form of this process, breaking down fatty acids into acetyl-CoA, which is then transported to the mitochondrial matrix through carnitine shuttle enzymes (carnitine palmitoyl transferases, CPT-1, and CPT-2). The deletion of CPT-1 increases HSC proliferation but decreases differentiation through blocking FAO, reducing acetyl-CoA synthesis, inhibiting the histone acetylation of differentiation-related factors in hematopoietic progenitor cells, and consequently reducing the cellular differentiation capacity [34]. The activation of FAO in HSCs is associated with high expressions of Cpt1a, Pml, and Ppard, suggesting the involvement of FAO in HSC regulation [35].

Peroxisome proliferator-activated receptor-delta (PPARδ) belongs to the ligand-activated receptor family and plays a role in fatty acid transport and oxidation regulation. PPARδ activity influences HSC function through various mechanisms. On one hand, the activation of the PPARδ-FAO pathway controls mitochondrial quality by enhancing the recruitment capacity of the Parkin protein and mitochondrial engulfment, promoting HSC expansion [36]. On the other hand, the PML-RARα fusion gene in early promyelocytic leukemia can induce asymmetric division in HSCs by activating PPARδ and promoting the FAO process [37]. When FAO is inhibited, HSCs tend to undergo symmetric division, generating differentiated hematopoietic progenitor cells, leading to a compromised self-renewal capacity and the depletion of the stem cell pool [35].

### 2.3. Purine and Amino Acid Metabolism

HSCs primarily remain in a dormant state by activating specific metabolic pathways such as glycolysis. However, during stress hematopoiesis, e.g., after BM transplantation, these metabolic activities undergo changes to adapt to alterations in the microenvironment [38]. The total concentration of over 20 amino acids in BM is much higher than in peripheral blood [20].

Daiki Karigane et al. found that p38α, a member of the p38-MAPK family, plays a crucial role in initiating the proliferation of hematopoietic stem and progenitor cells during stressed conditions in mice. Upon stimulation, p38-MAPK in HSPCs is immediately phosphorylated, promoting cell cycle entry. Mechanistically, the p38α signaling pathway increases the expression of inosine-5′-monophosphate dehydrogenase 2 (IMPDH2) in HSPCs, leading to changes in the levels of amino acids and purine-related metabolites [39].

Most cell growth and proliferation depend on endogenously synthesized asparagine. Asparagine synthesis and consumption are controlled by two reversible reactions. While synthesis occurs in the mitochondria through the action of glutamine and oxaloacetate catalyzed by glutamate-oxaloacetate transaminase 2 (GOT2), consumption occurs in the cytoplasm through the action of glutamate-oxaloacetate transaminase 1 (GOT1) [40,41]. A recent study reveals that HSCs are entirely reliant on the endogenous synthesis of asparagine for their regeneration [42]. Steady-state HSCs also limit protein synthesis for long-term maintenance. Under proliferative conditions, HSCs over-synthesize proteins, which leads to an increase in misfolded proteins and the formation of unfolded protein aggregates, resulting in protein toxicity stress and long-term damage to HSC function.

A direct link between metabolic changes and protein translation control in HSCs has been established. Steady-state HSCs exhibit a high amino acid (AA) catabolic metabolism to decrease cellular AA levels, thereby activating the GCN2-eIF2α-signaling axis, a protein synthesis inhibitory checkpoint, to suppress protein synthesis and maintain HSC function. When HSCs undergo proliferation, HSCs enhance mitochondrial OXPHOS to produce more energy, but reduce the AA catabolic metabolism to accumulate cellular AA. This deactivates the GCN2-eIF2α axis to increase protein synthesis and inhibit Src-mediated AKT activation to suppress mitochondrial OXPHOS in HSCs. The glycolytic metabolite nicotinamide riboside (NR), a precursor of NAD+, also accelerates the AA catabolic metabolism to activate GCN2 and maintain the long-term function of HSCs [43].

### 2.4. Other Metabolic Regulation Pathways in HSC

The distribution of nutrients in the BM microenvironment differs significantly from that in the bloodstream, shaping the unique metabolic characteristics of HSCs [44]. Nutrient metabolism is closely associated with the regulation of HSC homeostasis in BM. It is noteworthy that due to changes in contemporary lifestyles, such as nutritional excess and sedentary behavior, there is a significant increase in the risk of metabolic diseases.

Among them, hypercholesterolemia has garnered widespread attention globally due to its association with lethal cardiovascular complications. Compelling evidence suggests that cholesterol metabolism affects HSC function. A recent study revealed that apolipoprotein A-1 binding protein 2 (AIBP2, also known as Yjefn3) regulates vascular development in zebrafish, and knocking down Aibp2 increases the cholesterol content in zebrafish embryos [45]. Consistently, the cholesterol-lowering drug atorvastatin reduces cholesterol levels in *Aibp2*-deficient animals and significantly restores the expression of HSC marker genes. Cholesterol synthesis is regulated by transcription factor SREBP2. When the cellular cholesterol level decreases, SREBP2 translocates from the endoplasmic reticulum to the Golgi apparatus, undergoes a two-step protein cleavage to release the N-terminal active end, which enters the nucleus to activate the transcription of genes related to cholesterol biosynthesis. Gu et al. found that AIBP2-mediated cholesterol efflux activates SREBP2, which subsequently binds to the Notch1 promoter, transcriptionally regulating the activity of the Notch signaling pathway, thereby controlling the formation and differentiation of HSCs [46]. High cholesterol not only promotes the myeloid-biased differentiation of HSCs through activating the SLC38A9-mTOR signaling axis, but also enhances the iron death resistance of HSCs by upregulating SLC7A11/GPX4 expression and inhibiting iron autophagy, ultimately leading to the expansion of myeloid-biased HSCs [47].

Glutamine, synthesized from glutamate and ammonia catalyzed by glutamine synthetase (GS), is a precursor in the tricarboxylic acid cycle and the most abundant non-essential amino acid [48]. Glucose and the glutamine metabolism overlap in important ways. Glutamine transport is a rate-limiting step activated by mTOR, which upregulates Glut1, promoting glucose transport in the body [49]. HSCs depend heavily on glutamine metabolism during erythroid differentiation, which requires the ASCT2 glutamine transporter protein and active glutamine. Blocking the erythropoietin/EPO-stimulated pathway of HSC differentiation to monocytes leads to stress responses, such as hemolytic anemia. Under stress conditions, glutamine, and glucose act as fuel for nucleotide biosynthesis, regulating HSC lineage differentiation [50]. In fact, excessive glutamine synthesis has been linked to disease progression in acute myeloid leukemia [51]. It has been shown that the m6A-binding protein IGF2BP2 promotes AML development and leukemia stem cell (LSC) self-renewal in an m6A-dependent manner by regulating key targets in the glutamine metabolism pathway (such as MYC, GPT2, and SLC1A5). The inhibition of IGF2BP2 by a recently discovered small molecule compound (CWI1-2) shows promising anti-leukemia effects both in vitro and in vivo.

HSC-specific Sphingosine kinase 2 (Sphk2) catalyzes Sphingosine-1-phosphate (S1P), a multifunctional, small, signaling molecule that regulates various biological effects, including tumorigenesis, angiogenesis, blood cell migration, and erythrocyte differentiation [52]. Sphk2 inhibition effectively improves the anaerobic metabolism of HSCs, enhances the stemness and functional repair of HSCs, and therefore delays HSC aging [53]. Sphk2 also regulates two key molecules in the degradation process of HIF1α, namely, the hydroxylase modification enzyme PHD and ubiquitin ligase VHL [54]. It has been shown that Sphk2 deletion in HSCs leads to a massive accumulation of HIF1α, protecting the low oxygen glycolytic metabolic state of HSCs and maintaining their function [55]. By contrast, high expression of Sphk2 in aged HSCs leads to HIF1α degradation, therefore reduces reactivity to low oxygen conditions, and results in the failure in maintaining the anaerobic glycolytic metabolic state. These account, at least in part, for the functional decline of HSCs in aging.

## 3. Gut Microbiota and HSC Metabolism

Beyond the intrinsic metabolic regulation of HSCs, the gut microbiota is one of several extrinsic influences assuming a crucial role in sustaining HSC function within the body. A recent study unveiled the impact of gut microbiota on the hematopoietic function of animals. The administration of broad-spectrum antibiotics to mice results in the eradication of gut microbiota, ultimately leading to a reduction in the quantity of stem cells and progenitor cells in the BM, which are not induced by the toxic effects of antibiotics on hematopoietic cells but rather by the elimination of gut microbiota through antibiotic treatment [56]. Mechanistic investigations have identified that steady-state hematopoiesis relies upon contributions from MyD88, STAT1, and type 1 interferon signaling via the bacterial microbiome [57,58,59].

Dysbiosis or compositional changes in gut microbiota have been linked to stem cell aging due to dysregulations in metabolism, aberrant activations of the immune system, as well as promoting epigenetic instability [60]. The metabolic changes in aging stem cells contribute to the accumulation of mitochondrial damage associated with the imbalance between glycolysis and OXPHOS, and the accumulation of ROSs resulting in the depletion of stem cell pools [61]. However, further investigations are needed to understand whether fecal microbiota transplantation from young donors to old recipients restores HSC function to ameliorate the health span.

## 4. Metabolism Meets HSC Genomic Integrity

Cellular metabolism is intimately linked to the maintenance of genomic integrity, with metabolic cues influencing DDR pathways and vice versa [5]. In general, the DNA damage in HSCs is endogenous, majorly induced by reactive oxygen species, aldehydes, and replication stress [62].

ROSs produced during metabolism damage DNA mainly through the reaction of hydroxyl (OH) with purines, pyrimidines, or the sugars of the DNA backbone. One of the most frequent oxidative DNA lesions is 8-oxo-dG [63]. As the powerhouse of the cell, mitochondria are vulnerable to oxidative damage and represent the main source of ROSs. They are considered key tuners of ROS metabolism and buffering, whose dysfunction can progressively impact genome integrity [64]. ROSs can additionally create oxidative damage on other biomolecules, such as proteins and lipids. Lipid peroxidation is also an important endogenous source of DNA damage. While ROS production can be detrimental to cells, it can also be beneficial since some free radicals are required to stimulate cellular responses and upregulate antioxidant pathways [65].

Endogenous metabolites generate mutagenic DNA adducts, posing a critical threat to the genome. Acetaldehyde, a well-studied endogenous aldehyde, is highly toxic to cells, presumably resulting from the generation of intrastrand and DNA-protein crosslinks, which can further lead to DNA double-strand breaks (DSBs). The accumulation of DNA damage in cells promotes their self-degradation. Incorrect repairs can trigger the onset of malignant transformation. Aldehyde dehydrogenase 2 (ALDH2) is known to prevent the accumulation of acetaldehyde by efficiently oxidizing it to acetic acid. Surprisingly, ALDH2 deficiency is well-tolerated in the human body, possibly due to the additional protection provided by a major FA protein, FANCD2. Indeed, the genetic inactivation of ALDH2 and FANCD2 in mice results in cancer and hematopoietic abnormalities, suggesting that endogenous aldehyde metabolic defects are a pervasive source of DNA damage, impairing genomic integrity in HSCs [66].

DNA methylation plays a critical role in maintaining genome integrity [67,68]. Stem cells demand a balanced histone methylation status to maintain their functionality [69]. It is known that metabolism tightly regulates DNA methylation [69], which is generally linked to transcriptional repression and established by DNA methyltransferases (DNMTs) on CpG islands. Conversely, DNA demethylation is catalyzed by TET (ten-eleven translocation) enzymes, which oxidize 5-methyl-cytosine (5mC) to 5-hydroxymethyl-cytosine (5hmC). Epigenetic regulatory enzymes are dependent on the metabolite availability to facilitate DNA- and histone-modifying reactions [69]. For example, α-ketoglutarate (αKG)-dependent dioxygenases, TETs that have been linked to epigenetic modifications, genome stability and DNA repair, are known to play a critical role in normal and pathological hematopoiesis. The activity of TETs is dependent on the αKG metabolites mainly produced in mitochondria by different metabolic pathways, including those that are a key intermediate in the TCA cycle for energy metabolism [70]. Additionally, DSB sensor, SIRT6 (Sirtuin6), a critical epigenetic regulator of glucose metabolism [71], negatively regulates HIF-1α-dependent transcription by deacetylating H3K9Ac at the promoter of GLUT1, lactate dehydrogenase A (LDHA), and PDH kinase 1 (PDHK1), thereby elevating glucose uptake and glycolysis even under normoxic conditions [72].

The activity of chromatin-modifying enzymes is highly dependent on essential metabolic cofactors [69]. For instance, acetyl-CoA is a donor of acetyl groups for histone acetylation [73]. In contrast, NAD^+^-dependent sirtuin (SIRT) and Zn^2+^-dependent histone deacetylases (HDACs) remove these groups from histones [74], thereby leading to chromatin compaction and transcriptional silencing. SIRTs are regulated by the NAD^+^/NADH ratio in the cell which directly links their activity to the cellular redox status [74]. Nuclear acetyl-CoA is generated through acyl-CoA-synthetase short-chain family member 2 (ACSS2), which generates acetyl-CoA from acetate, and ATP-citrate lyase (ACLY), which produces acetyl-CoA from citrate. In addition, the pyruvate dehydrogenase complex translocates to the nucleus during mitochondrial stress and generates acetyl-CoA from pyruvate [75]. Chromatin is not only passively impacted by metabolism and DNA damage, but also actively regulates metabolism and the DNA damage response [76]. For instance, Dpy30, a common core subunit of the SET1/MLL complexes that catalyze H3K4 methylation, is known to sustain long-term HSC self-renewal and enable their differentiation [77,78], through regulating both glycolysis and mitochondrial activities.

## 5. Fanconi Anemia, Genomic Integrity, and HSC Metabolism

DNA damage repair is a complex signal transduction network that is required for preserving the integrity of the genome and for ensuring its accurate transmission through generations. To counteract DNA damage, DDR machinery orchestrates DNA damage checkpoint activation and facilitates the removal of DNA lesions. Unrepaired damage results in cellular senescence or apoptosis while erroneously repaired DNA lesions can lead to mutations [79,80,81]. The dysregulation of DDR and repair systems can cause human disorders associated with cancer susceptibility, accelerated aging, and developmental abnormalities [82]. Given the enormous regenerative potential coupled with the lifetime persistence of hematopoietic stem cells in the body, tight control of HSC genome stability is demanded. In fact, DDR has been considered to be an evolutionary checkpoint between blood regeneration and leukemia suppression [83]. Failure to accurately repair DNA damage in HSCs is associated with bone marrow failure and leukemogenesis [83].

The repair machinery for DNA damage in HSCs has its own characteristics [62]. For example, the FA/BRCA (breast cancer susceptibility gene) pathway is particularly important, while the base excision repair (BER) pathway is less involved in the hematopoietic system. HSCs also prefer to utilize the classical non-homologous end-joining (NHEJ) pathway, which is essential for V(D)J rearrangement in developing lymphocytes and is involved in DSB repair to maintain genomic stability in the long-term quiescent state.

### 5.1. FA, DNA Damage Repair, and HSC Defects

Fanconi anemia is an inherited disorder associated with BM failure and leukemia [84,85,86]. It is genetically heterogeneous, with at least 23 complementation groups identified thus far. Among them, the genes encoding the groups A-W (FANCA-FANCW) have been cloned [66,87,88,89,90,91,92,93]. At the molecular level, the DNA damage repair-based FA pathway has been established, in which eight of the FA proteins (namely FANCA, B, C, E, F, G, L, and M) form the FA core complex (Figure 2). The core complex is required for the mono-ubiquitination of two downstream FA proteins, FANCD2 and FANCI, which then recruit other downstream FA proteins and DDR/repair factors to nuclear loci containing damaged DNA, and consequently influence important cellular processes such as DNA replication, cell-cycle control, and DNA damage response and repair [94,95]. As is well acknowledged, DDR deficiency resulting from the loss of FA proteins is considered to be the fundamental cause of BM failure and the predisposition to AML in FA.

The two most important clinical hallmarks of FA are BM failure and the progression to leukemia caused by HSC depletion and malignant transformation. FA commonly progresses from BM failure to a pre-leukemic myelodysplastic syndrome (MDS) stage and finally evolves to acute myeloid leukemia (AML). In fact, FA is among the most common BM failure syndromes and childhood leukemias seen in BM transplantation clinics. The complications of BM failure are the major causes of the morbidity and mortality of FA, and if left untreated, the majority of FA patients die from BM failure [84,85,86]. As a bona-fide HSC disease, FA represents a unique disease model for studying the mechanisms of BM failure and leukemogenesis.

### 5.2. Metabolic Alterations in FA HSCs

Emerging evidence suggests that DDR deficiency leads to a distinct metabolic reprogramming of HSCs. For example, HSCs from DDR deficient FA patients or mice exhibit significant metabolic reprogramming to maintain their functionality. Our recent study reveals that FA HSCs exhibit a heightened dependence on OXPHOS and undergo a rapid switch from glycolysis to OXPHOS under oxidative stress to cope with oxidative DNA damage. Mechanistically, the tumor suppressor p53 functions as the key master regulator mediating this transition. p53 regulates energy metabolism at the glycolytic and OXPHOS steps via the transcriptional regulation of its downstream genes, such as the synthesis of SCO2, a member of the COX-2 assembly involved in the electron-transport chain [7]. Knockdown SCO2 significantly reduces the oxygen consumption and hematopoietic reconstitution capacity of *Fanca*-deficient HSCs (Figure 3, route 1). In addition, p53 also inhibits glycolysis by negatively regulating the TP53-induced glycolysis regulator (TIGAR; Figure 3, route 2) [8]. Interestingly, the inhibition of glycolysis is specific to HSC compartment but not in the less primitive MPPs (Lin^−^Sca1^+^c-kit^+^CD150^-^CD48^−^), HPCs (Lin^−^Sca1^+^c-kit^+^CD150^−^CD48^+^ and Lin^-^Sca1^+^c-kit^+^CD150^+^CD48^+^), or Lin^+^ cells. This reflects the significant metabolic differences between FA HSCs and the differentiated progenitors, suggesting the shift in energy metabolism may be a major factor leading to hematopoietic abnormalities, and even the primary cause of BM failure and leukemia in FA [25].

Another member of the PPAR family, PPARγ, is known as a negative regulator of HSCs [96]. Our recent studies suggest that the loss of HES1 (hairy and enhancer of split 1) activates the FAO mediated by PPARγ, resulting in the exhaustion of FA HSCs under transplantation stress [9]. Metabolic pathways provide energy and building blocks for other factors functioning on DDR [97]. The inhibition of PPARγ enhances the DDR in FA HSCs, thereby restoring their hematopoietic reconstitution function (Figure 4A). Mechanistically, FANCD2 collaborates with the HES1 to suppress inflammation-induced PPARγ, contributing to the maintenance of HSCs. This program involves the upregulation of genes related to FAO and OXPHOS, showcasing how the loss of FANCD2 or HES1 amplifies both basal and inflammation-primed FAO (Figure 4B) [10]. These studies demonstrate that members of the PPAR family could serve as novel therapeutic targets for improving HSC function and treating hematologic diseases.

### 5.3. FA and Mitochondria Dysfunction

While many studies point to an essential role for the FA pathway in DDR and genome maintenance, compelling evidence suggests that FA deficiency also causes mitochondrion dysfunction [98,99], which may contribute to the pathogenesis of BM failure and leukemia progression. FA cells exhibit abnormalities in mitochondrial metabolism, Ca2^+^ homeostasis, and gene expression [98,100,101,102]. In addition, mitochondrial defects are also present in FA mice and patients [98,99,103,104]. FA deficiency alters mitochondrial morphology, causes mitochondrial complex defects, and decreases mitochondrial membrane potential and ATP production [102,105,106]. Some of the FA proteins are required for removing damaged mitochondria and reducing mitochondrial ROSs [107]. Furthermore, the utility of antioxidants shows in vivo protective effects against the onset of malignancies and BM failure in FA knockout mice [108,109]. By using a novel Flag- and hemagglutinin-tagged *Fancd2* knock-in mouse model, our recent studies identify the interaction between Fancd2 localization in the mitochondrion and the association with the nucleoid complex components Atad3 and Tufm [110], thus providing a molecular link between FA and mitochondrial homeostasis for the first time (Figure 5).

## 6. Recent Technological Advances in HSC Metabolism

Many routine metabolic tools are not suitable for delineating the metabolic properties of HSCs due to the scarcity of HSCs. Recently, several new/optimized techniques have been developed to fulfill the needs of the delineation of the complicated metabolic networks in HSCs. For example, glycolysis and mitochondrial respiration in HSCs can be measured as the extracellular acidification rate (ECAR) and the oxygen consumption rate (OCAR) using a Seahorse XF analyzer [35]. Newly developed LC-MS makes it possible to measure various intermediate metabolites with a small number of HSCs [111]. Catabolism (anabolism) is a redox bioreaction that involves the electron transfer between different substrates mediated by specific coenzymes, such as nicotinamide adenine dinucleotide and nicotinamide adenine dinucleotide phosphate (NADP+), and results in energy synthesis and release. Recent studies developed genetically encoded fluorescent sensors that can be used to monitor the ratio of NAD^+^/NADH or NADP^+^/NADPH changes in live cells [112,113] and to evaluate the glucose or amino acid changes in HSCs or leukemia cells [114,115]. More recently, two low-input liquid chromatography-tandem mass spectrometry (LC-MS/MS) approaches, either detecting semi-polar metabolites such as TCA cycle metabolites and amino acids, or measuring polar lipids, bile acids, and retinoids, have been established. Researchers successfully identified 160 metabolites, including phosphatidylcholine, glycolytic and lipid intermediates in HSPCs [116] and purified HSC populations [117]. These new metabolic techniques provide potent tools to precisely evaluate the subtle dynamic metabolic changes in HSCs or other types of stem cells with high sensitivities.

Due to some technical difficulties, such as low amounts of total mRNA per cell in relative quiescent and less active HSCs, the metabolic heterogeneity of HSCs has never been directly measured [118]. A recent study established a new platform for high-throughput single-cell metabolomics (hi-scMet), which allows for the detection of over 100 features from single cells by combining flow cytometry cell isolation and nanoparticle-enhanced laser desorption/ionization mass spectrometry [119]. By mapping the single-cell metabolomes of all hematopoietic cell subpopulations with different division times, hi-scMet revealed a progressive activation of the oxidative pentose phosphate pathway (OxiPPP) from dormant to active HSCs. While further investigation continues to be needed, this novel platform highlights the potential of dissecting the metabolic heterogeneity of immunophenotypically defined cell populations in physiological or different disease conditions.

The intricate relationship between DDR deficiency and HSC metabolism unveils a fascinating landscape where cellular energetics and genomic stability converge. With progressed studies on the role of FA genes in non-FA human cancer development and progression, FA has stood out to be a unique biological model system to study cancer etiology and treatment [120,121,122]. Vice versa, FA HSCs are also becoming a distinct biological model system to explore the interplays between metabolism and DDR in depth. This review provides a comprehensive overview of the current understanding of this interplay, highlighting the potential for therapeutic interventions that harness the metabolic prowess of HSCs to enhance DNA repair and maintain cellular homeostasis. As the research unfolds, the metabolic crossroads of HSCs and DDR promise to be fertile grounds for discoveries that could reshape our approaches to stem cell biology and regenerative medicine.

## Figures and Tables

**Figure 1 cells-13-00733-f001:**
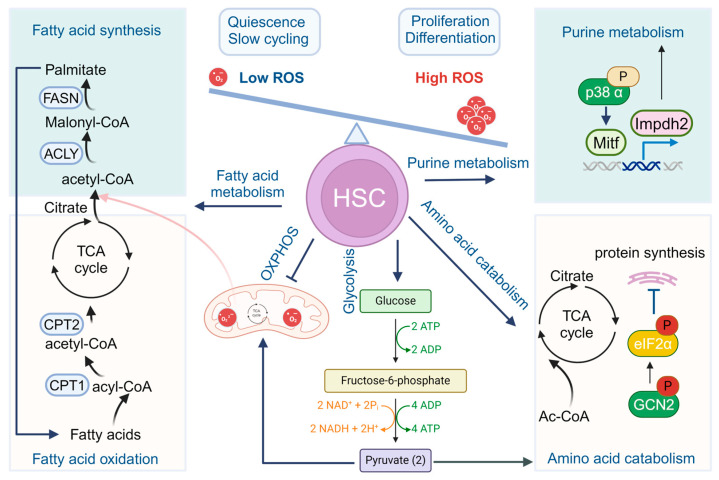
Metabolism in HSCs. Quiescent HSCs preferably utilize glycolysis rather than OXPHOS for energy production. During HSC differentiation and maturation, a rapid switch from glycolysis to mitochondrial OXPHOS and ATP generation occurs. Other metabolism pathways, including fatty acid metabolism, purine and amino acid metabolism, and cholesterol metabolism also participate to fine-tune the stemness and cell fate commitments of HSCs.

**Figure 2 cells-13-00733-f002:**
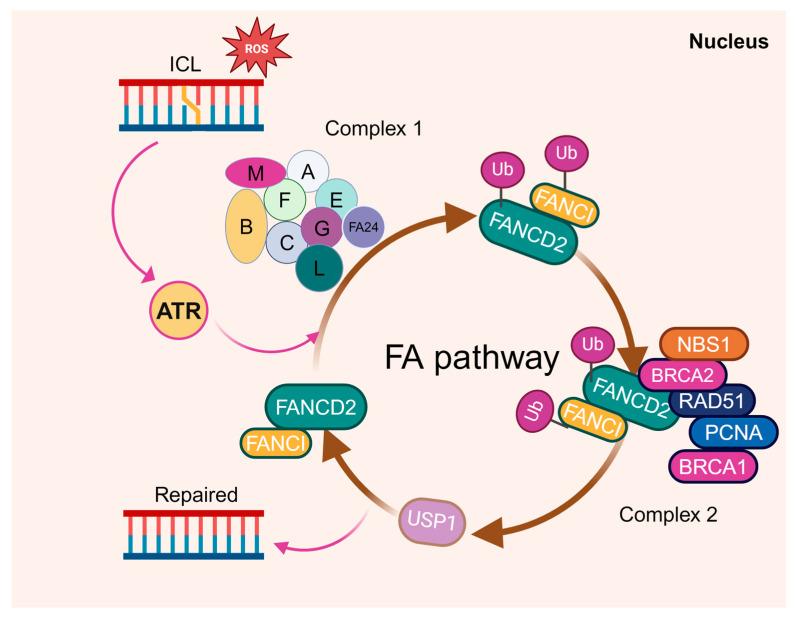
The FA proteins function in DNA damage repair. Eight of the FA proteins (namely FANCA, B, C, E, F, G, L, and M) form the FA core complex to mono-ubiquitinate two downstream FA proteins, FANCD2 and FANCI, upon DNA damage. The two then form a dimer to recruit other downstream FA proteins, such as FANCD1, FANCJ, and FANCN, to the damaged DNA and influence the DNA replication, cell-cycle control, and DNA repair processes.

**Figure 3 cells-13-00733-f003:**
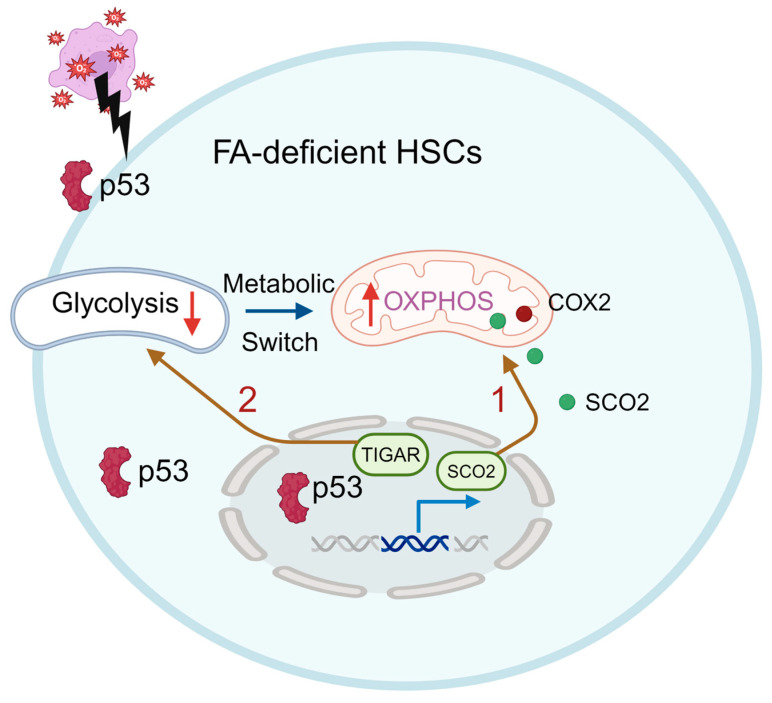
p53 participates in the FA HSC metabolism. p53 regulates energy metabolism at the glycolytic and OXPHOS steps via the transcriptional regulation of its downstream genes, such as the synthesis of cytochrome c oxidase (SCO2), and inhibits glycolysis by negatively regulating the TP53-induced glycolysis regulator (TIGAR). FA HSCs are more dependent on OXPHOS and undergo a glycolysis-to-OXPHOS switch mediated by SCO2 under oxidative stress (Route 1). p53-TIGAR metabolic axis-mediated glycolytic suppression plays a compensatory role in attenuating DNA damage and proliferative exhaustion in FA HSCs (Route 2).

**Figure 4 cells-13-00733-f004:**
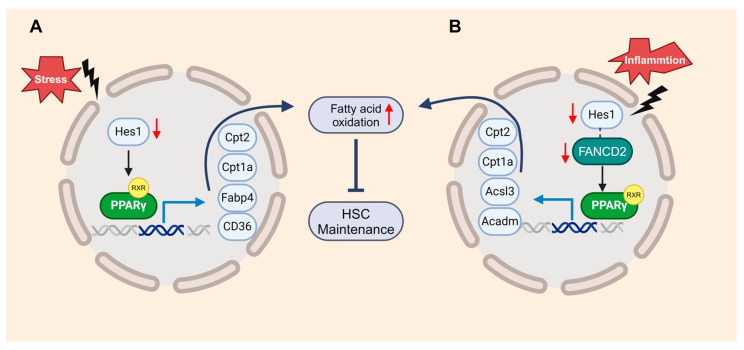
FANCD2 and HES1 act in concert to suppress inflammation-induced PPARγ to prevent HSC exhaustion through restricting fatty acid oxidation (FAO). (**A**). Loss of *Hes1* deregulates genes in PPARγ signaling and FAO, thereby augment FAO in HSPCs. (**B**). A novel FANCD2/HES1/PPARγ axis constitutes a key component of immunemetabolic regulation, connection inflammation, cellular metabolism and HSC function.

**Figure 5 cells-13-00733-f005:**
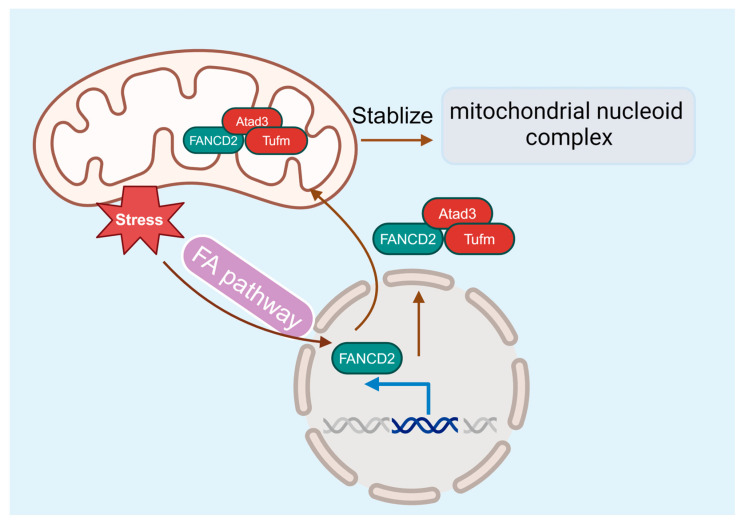
The FA pathway and mitochondria. FANCD2 localizes in the mitochondrion and associates with the nucleoid complex components Atad3 and Tufm, linking the FA pathway and mitochondrial homeostasis.

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
