# Peer review of "Crosstalk between DNA Damage Repair and Metabolic Regulation in Hematopoietic Stem Cells"

_cells, 2024, doi:10.3390/cells13090733_

Round 1
Reviewer 1 Report
Comments and Suggestions for Authors
In this review, J. Xu et al. analyze the relationships between metabolic regulation and DNA repair in hematopoietic stem or progenitor cells.
It is comprehensive and well organized. The metabolic part is very complete and explains the changes that occur between quiescent HSCs and proliferating progenitors. However, I was a little disappointed when it came to DNA repair. Indeed, the link between metabolism and DNA repair is succinctly described, focusing mainly if not exclusively on the Fanconi anemia pathway. There are many other connections between metabolic and DNA repair pathways. As na exmaple, pathways involving α-ketoglutarate (αKG)-dependent dioxygenases such as TET proteins, which in turn are linked to epigenetic modifications, genome stability and DNA repair, play a very important role in normal and payhological hematopoiesis. The activity of TETs is dependent on α-KG metabolite which is produced by different different metabolic pathways, including as a key intermediate in the TCA cycle for energy metabolism.
As a result, it would be preferable to change the title of the review to make it clear that it will focus on the connections between FA pathway DNA repair deficiency and metabolic regulation.
The term "intricacies" itself is a misleading. It would be preferable to use crosstalks, or connections
Author Response
To address the reviewer’s concern on narrow focus on FA, we have extensively expanded “Metabolism meets HSC genomic integrity” section to include other factors involved in maintaining genome integrity. We also changed the title to “Crosstalk between DNA damage repair and metabolic regulation in hematopoietic stem cells”.
changes were highlighted in blue
Reviewer 2 Report
Comments and Suggestions for Authors
The authors in this manuscript reviewed the metabolic contribute on the hematopoietic stem cells. The manuscript is clear and the main topic is well resolved. I think that could be very interesting If the authors add some more information about the last sc-rnaseq focusing on metabolism that are present in literature. This could be emphasize this review in order of concept but also regarding the novelty.
The authors could also add additional information regarding metabolomic data or other data that could better highlight the role of metabolism in this field of study.
Author Response
We have added the discussing on new scRNA-seq analysis platform on page 11.
changes were highlighted in blue.